# Experimental study on the various varieties of photovoltaic panels (PVs) cooling systems to increase their electrical efficiency

Ali Basem[1], Azfarizal Mukhtar[2], Zakaria Mohamed Salem Elbarbary[3]*, Farruh Atamurotov[4,5,6], Natei Ermias Benti[7]*

1 Air Conditioning Engineering Department, Faculty of Engineering, Warith Al-Anbiyaa University, Karbala, Iraq, 2 Institute of Sustainable Energy, Putrajaya Campus, Universiti Tenaga Nasional, Kajang, Malaysia, 3 Electrical Engineering Department, College of Engineering, King Khalid University, Abha, Saudi Arabia, 4 New Uzbekistan University, Tashkent, Uzbekistan, 5 University of Public Safety of the Republic of Uzbekistan, Tashkent Region, Uzbekistan, 6 University of Tashkent for Applied Sciences, Tashkent, Uzbekistan, 7 Computational Data Science Program, College of Computational and Natural Science, Addis Ababa University, Addis Ababa, Ethiopia

* albrbry@kku.edu.sa (ZMSE); natei.ermias@aau.edu.et (NEB)

**Data Availability Statement:** All relevant data are within the manuscript.

**Funding:** The authors extend their appreciation to the Deanship of Scientific Research at King Khalid

## Abstract

This study investigates the impact of cooling methods on the electrical efficiency of photovoltaic panels (PVs). The efficiency of four cooling techniques is experimentally analyzed. The most effective approach is identified as water-spray cooling on the front surface of PVs, which increases efficiency by 3.9% compared to the case without cooling. The results show that water-spray cooling raises the PV's temperature to 41°C, while improving its average daytime efficiency to 22%. Air-cooling, water-cooling in the tubes behind the PV, and aluminum oxide-water nanofluid cooling in the tubes behind the PV improve efficiency by 1.1%, 1.9%, and 2.7%, respectively. The findings highlight the potential of water-spray cooling as a cost-effective and efficient method to enhance PV efficiency and contribute to the global effort towards renewable energy.

## 1. Introduction

The increase in the world population in the modern era has led to a notable rise in the demand for electricity [1]. Fossil fuel-based energy sources, which have been shown to have serious negative consequences on the environment and human health, provide the bulk of the world's electrical energy. The release of greenhouse gases into the atmosphere causes these harmful effects, which in turn cause climate change and global warming. Many academics have turned to renewable energy as a possible means of addressing the growing human demand for electricity and the related issues brought on by shifting climatic patterns and environmental circumstances [2, 3]. Most people agree that one of the easiest energy sources to use to meet today's needs is solar energy [4, 5]. Due to the sun's abundant presence on Earth's surface, this precious resource serves as a perfect solution to the problem of electricity and energy shortages [6]. In addition to its economic advantages, the use of solar energy on Earth can help reduce

University under for funding this work through General Research Project under Grant number (RGP.2/33/44) The funders had no role in study design, data collection and analysis, decision to publish, or preparation of the manuscript.

**Competing interests:** The authors have declared that no competing interests exist.

pollution, which is a major problem, and help preserve the planet [7–10]. In the Middle East, and especially in Saudi Arabia, solar energy is of a high caliber. In some nations, the yearly solar radiation surpasses 2100 (Kw.h∕m^2) [11, 12].

One approach to harnessing solar energy for electricity generation involves the utilization of photovoltaic panels (PVs) [13–15]. These panels facilitate the direct conversion of solar radiation into electrical energy. The solar PV system is widely regarded as a highly appealing choice for electricity generation when compared to other energy systems. This is primarily attributed to its advantageous characteristics, such as its cost-effective maintenance, simplified installation process, and impressive efficiency [16]. Hence, in light of the aforementioned benefits, it is imperative to implement measures aimed at enhancing the efficacy of PV systems [17].

In the conducted research, the investigators demonstrated that with every incremental rise in temperature, the PV system experiences a reduction in electrical efficiency ranging from approximately 0.4% to 0.5% [18]. The decline in efficiency can be attributed to the rise in saturation current and decline in voltage of PV cells under elevated temperatures [17]. Hence, the regulation and management of the temperature in the PV system can have an impact on enhancing its efficiency and overall performance [19–22]. Numerous findings have been derived from studies examining the impact of temperature fluctuations and solar radiation on the efficiency of PV systems [23]. Several studies have indicated that a temperature rise can result in a reduction in the efficacy of PV systems [24]. This is explained by the inverse connection between PV cell voltage and temperature, which states that a rise in temperature causes a drop in cell voltage. Consequently, this decline in voltage adversely affects the production of electricity. Moreover, empirical studies have demonstrated that augmenting solar radiation can result in heightened efficacy of PV systems [25]. This is explained by the fact that solar radiation and the current produced by photovoltaic cells are positively correlated, which increases the amount of power produced. Temperature fluctuations and solar radiation are two crucial variables that significantly impact the efficacy of PV systems [26].

Numerous studies have demonstrated that the implementation of techniques aimed at managing the temperature of PV systems can effectively enhance their efficiency [27–29]. According to Jailany et al. [30], enhancing the efficiency and output power of the PV system is primarily contingent upon the reduction of the PV surface temperature. Tiwari et al. [31] conducted a study to investigate the impact of ambient temperature on the efficiency of PVs. The individuals conducted their research during the summer period. The researchers' findings indicated that the PV system exhibited diminished efficiency during daylight hours as a result of elevated ambient air temperatures. Wang et al. [32] examined the performance of solar cells immersed in various liquids under simulated sunlight. They found the electrical properties of the liquids, rather than the optical properties, had the greater impact on solar cell efficiency. Bare solar cells in non-polar silicon oil showed the best performance. Accelerated aging tests indicated the silicon oil has good stability. The authors suggest this work supports using liquid immersion to cool concentrated photovoltaic systems.

Several researchers have proposed several cooling technologies with the aim of controlling the temperature of photovoltaic panels [33–36]. Sheikholeslami et al. [37] proposed a heat transfer tube design method for photovoltaic systems. They found that an 8-lobed tube outperformed a circular one, and copper fins improved efficiency. Using graphene nanoplatelets as a heat transfer fluid enhanced electrical performance by 5.8%. The long-term performance of distinct cooling technologies for photovoltaic modules was assessed by Bevilacqua et al. [38]. These technologies included forced ventilation and spray cooling, which were applied to the back surface of the PV modules. They discovered that the cooling systems were capable of reducing the back temperature of the PV modules by up to 26.4°C on sunny days and

enhancing the uniformity of temperature distribution. Cuce et al. [39] examined the influence of passive cooling on the effectiveness of PV cells. The researchers utilized an aluminum heat sink to efficiently disperse any surplus heat produced during the operation of the cells. The investigation entailed conducting trials with different ambient temperature values and sun radiation intensities. The study's findings suggest that the proposed cooling strategy has been shown to improve energy conversion efficiency, exergy, and cell power by around 20% when exposed to an irradiation of 800 W/m2. Stalin and his colleagues [40] suggested using nano-PCM to efficiently control the temperature conditions of PVs. The researchers have shown that employing this technique leads to a significant decrease in the temperature of the PV surface, which in turn improves the electrical efficiency. In a different work, Said et al. [41] provided a hybrid cooling model for photovoltaic systems. This model combines the usage of an active cooling system with the passive cooling approach using phase change material (PCM) and water spraying. In a recent experimental work conducted by Patil et al. [42], the authors investigated the application of air as a cooling mechanism for photovoltaic systems. The research findings showcased the capacity to improve the efficiency of photovoltaic systems by lowering their temperature using this way. The authors conducted a distinct investigation where they examined the computational fluid dynamics for the process of cooling photovoltaic systems using air. According to their research, the addition of an air-conditioning system improves the overall efficiency of the system compared to situations where cooling is not present [43]. Hussien et al. [44] conducted an experimental-numerical study to analyze the effects of a forced air-cooling model on the efficiency of a photovoltaic system. The study's findings revealed that the use of this cooling technique led to a decrease in the temperature of the photovoltaic system, resulting in an improvement in its efficiency by up to 1.34%. An experimental investigation was carried out by Nabil et al. [45] to look into various cooling options for solar panels. By conducting a comparative examination of several cooling approaches, it was proven that applying water spray over the photovoltaic system results in a more effective drop in temperature compared to other ways. Furthermore, the application of the water spraying approach demonstrates a significant improvement in the electrical efficiency of the photovoltaic system in comparison to other ways. Bevilacqua et al. [46] proposed a novel one-dimensional thermal model that accounts for the spray cooling phenomenon on the back surface of photovoltaic panels. The model, based on the energy balance method and a finite difference approach, was validated and showed high accuracy in predicting the back surface temperature and the electrical power output. Sheikholeslami et al. [47] designed a concentrated photovoltaic-thermal (CPVT) system with linear Fresnel concentrators and a thermoelectric generator (TEG) to increase productivity. They used alumina-water nanofluid jets to cool the system and investigated the impact of inlet temperature and velocity on performance. Odeh et al. [48] proposed a solar-water pumping system to improve the efficiency and reduce the thermal degradation of a PV module. The system consists of a PV module, a submersible water pump, and a water storage tank. Cooling of the PV panel is achieved by introducing a water trickling configuration on the upper surface of the panel.

Rostami et al. [34] proposed a novel method that employs atomized CuO nanofluid and high-frequency ultrasonic waves for the purpose of cooling PV modules. The incorporation of CuO nanoparticles in the study was driven by their unique characteristics and remarkable conductivity. The results suggest that using the proposed approach leads to a substantial increase in cooling capacity, ranging from 2.75% to 57.25%. Furthermore, there is a proportional rise in the maximum power, varying from 3.4% to 51.2%. These enhancements depend on the individual factors of vapor flow rate or ultrasonic power. Chandrasekar et al. [49] conducted a study to examine the use of wicks placed on the back of PV modules to improve heat transmission and increase cooling efficiency in a cost-effective way. The study's findings indicate that

the PV reached a temperature of 65°C without any cooling. Still, after adding a cooling wick soaked in water, the temperature dropped to 45°C. To reach a 20°C temperature drop, Idoko et al. [50] used surface water-cooling and mounted an aluminum heat sink to the rear of PV modules. The utilized cooling system demonstrated the ability to improve the efficiency of the module by at least 3%. Sheikholeslami et al. [51] designed a concentrated photovoltaic thermal (PVT) system to increase electricity generation and heat saving. They used nanofluid and paraffin containers with MWCNT nanoparticles and tested different fin configurations to optimize performance.

Renewable energy systems play a crucial role in achieving sustainable development goals today. In this context, investigating and improving the performance of these systems is of particular importance. While the electrical efficiency of PV panels is the main focus of this study, other aspects such as environmental and energy assessment should also be considered to fully realize the potential of renewable energy technologies [51].

For example, the integration of PV-thermal systems with reflectors and nanofluid filters has shown potential to enhance the overall efficiency of these systems. Similarly, the use of nanofluid spectral splitters in concentrated solar PV-thermal systems can lead to increased efficiency by optimizing the utilization of the solar spectrum. These studies demonstrate that the development and optimization of renewable energy systems should be examined from various dimensions, including not only the electrical performance but also the thermal and environmental aspects [52].

In line with this multifaceted approach, the present research focuses on investigating the effect of different cooling methods on the efficiency of PV panels, which is a crucial factor in improving the overall performance of PV systems. Numerous studies have indicated that the implementation of a cooling system for PVs can enhance their overall performance [52]. In this study, four cooling methods were evaluated, including forced air-cooling, water-cooling through tubes, nanofluid-based cooling, and water-spraying on the front of the PV. The results of this comparative analysis highlight the significance of adopting an effective cooling method, with water-spray cooling emerging as the most promising approach for enhancing PV efficiency. This investigation provides valuable insights into the potential of cooling systems to improve the performance of PVs, ultimately contributing to the development of sustainable energy solutions.

## 2. Methodology

### 2.1. Experimental model

The thermal management of PVs is a critical determinant of their overall efficiency and optimal operational capabilities. The efficiency of PVs decreases when they experience an increase in temperature during operation. Hence, the implementation of adequate cooling measures can yield advantageous outcomes in terms of both PV performance and longevity. This section presents a discussion on the cooling techniques employed for PVs to enhance their operational efficiency. In this context, four distinct instances of these cooling techniques have been constructed and examined. Methods include the following:

**2.1.1. Air-cooling.** The utilization of air as a cooling agent is employed in this particular method. In order to facilitate the cooling of PVs, an aluminum plate is strategically positioned beneath the PVs to allow for the passage of air. Silicone adhesives are employed to affix the aluminum plate and mitigate the decline in thermal conductivity. Additionally, a fan is employed to facilitate the flow of air over these plates, thereby enhancing the heat transfer rate through forced convection. Additionally, Fig 1 presents a visual representation of the aforementioned system. The technical specifications of the various components comprising this system are detailed in Table 1.

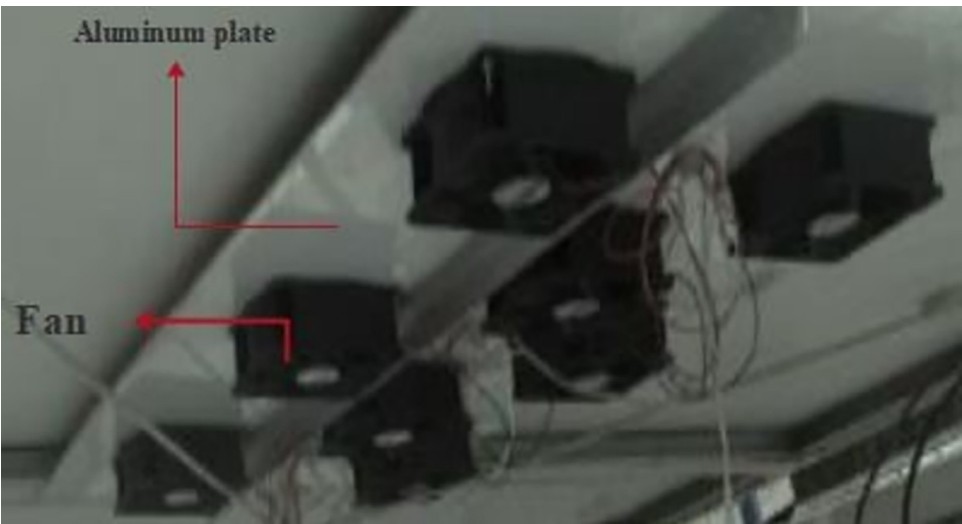

**Fig 1. A view of the air-cooling method installation under the PV.**

**2.1.2. Liquid cooling from the back.** This approach involves the utilization of fluid-filled tubes that are connected to the rear side of the PVs. A model from the Alizadeh et al. [53] study has been used to implement this system. The fluid is propelled through the pipes via a pump. The fluid contained within the tubes serves as a coolant medium, facilitating the absorption of thermal energy from the PVs and subsequently facilitating its dissipation from the system. This study examines the characteristics of two distinct fluids, namely water and aluminum oxide-water nanofluid, in isolation within the system. The model proposed by Chamkha et al. [54] was utilized as a guide for the nanofluid experiment. Fig 2 depicts a visual representation of the pipes affixed to the rear side of the PVs. The specifications of the components related to this system are presented in Table 2.

**2.1.3. Water-cooling from the front surface.** In this approach, a fine mist of water is applied to the frontal surface of the PV. Through the process of evaporation, the water effectively absorbs heat from the PVs, thereby maintaining a cool temperature for the PVs. The operational procedure employed in this methodology is characterized by its straightforwardness and minimal reliance on equipment. The configuration of the cooling method is depicted in Fig 3. The technical details about the PV utilized in this methodology are outlined in Table 2.

**Table 1. Specifications of system components with forced air-cooling.**

| Components | Specifications |
|---|---|
| Photovoltaic panel | dimensions: 1530 × 760 mm<br>Weight: 14 kg<br>Number of cells: 63<br>Features: Monocrystal<br>Connector type: MC4 |
| Aluminum plate | dimensions: 1400×600×2 mm |
| Silicone thermal paste | Silicone glue RTV-50ml<br>Model: MT-704 Maxtor brand<br>TCC: 0.95W/m-k |
| Fan | 8× 12 V and 0.3 A |

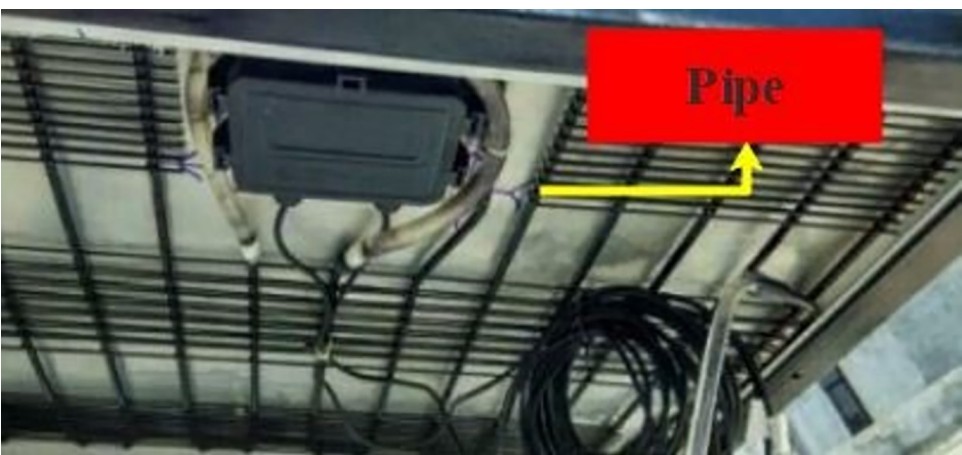

**Fig 2. Pipes affixed to the underside of the PV to transport the cooling fluid.**

## 2.2. Measuring equipment

To assess the constructed systems, it is imperative to measure certain parameters throughout the day. The intensity of solar radiation is a crucial parameter that significantly influences the efficiency of PVs. The measurement and recording of this parameter are facilitated by a Solar power meter. Additional parameters that hold significance for evaluation encompass the environmental temperature and the temperature of the PV. Previous research conducted by other scholars has demonstrated a substantial correlation between the efficiency of PVs and their respective temperatures [55]. To fulfill the intended objective, a pair of NTC100 temperature sensors have been chosen to accurately gauge the temperature of both the PV surface and the ambient surroundings. To quantify the power generated by the PV, two multimeters were employed to assess the voltage and current intensities. Table 3 presents the specifications and measurement accuracy of each piece of equipment. The images of each measuring equipment are shown in the last column of Table 3.

The temperature sensors were placed at the center of the PV panel, where the maximum temperature is typically observed. This is because the center of the panel is the hottest part of

**Table 2. Specifications for components of PVs with liquid cooling from the rear.**

| Components | Specifications | |
|---|---|---|
| Photovoltaic panel | dimensions: 1530 × 760 mm<br>Weight: 14 kg<br>Number of cells: 63<br>Features: Monocrystal<br>Connector type: MC4 | |
| Pipe | Material: Copper<br>Inner diameter: 10 mm<br>Outer diameter: 12 mm<br>Total length: 11.5 m | |
| Pump | Power: 25 watts<br>Voltage: 12 volts<br>Flow rate: 10.5 liters per minute | |
| Cooling fluid | Fluid1 | Water |
| | Fluid2 | Aluminum oxide-water nanofluid |

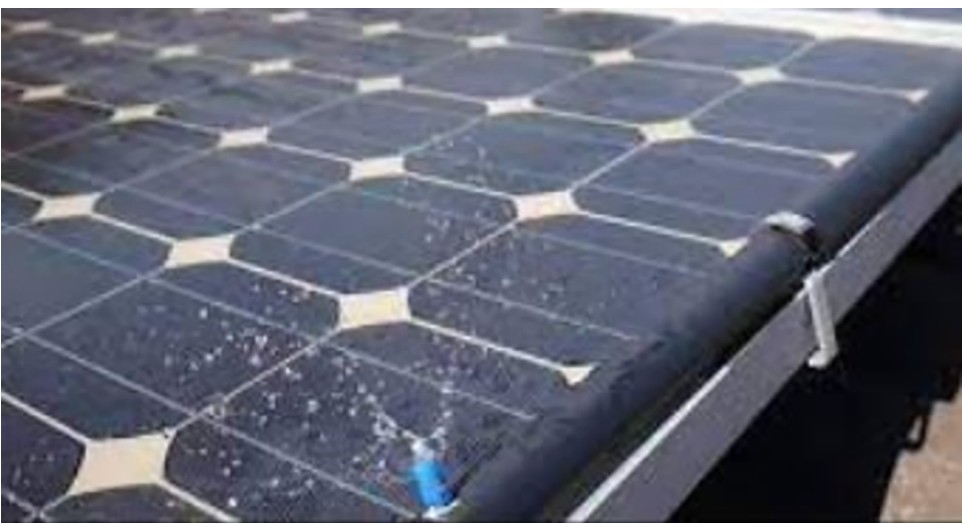

**Fig 3. Water spraying method for cooling the PV surface.**

the PV, and controlling the temperature of this region, which is critical, can have a significant impact on the overall efficiency of the PV (see Fig 4).

## 2.3. Experimental methodology

To enhance the efficiency of PVs, an investigation was conducted to examine various cooling methods. A representative sample was created for each method and subsequently analyzed. The methods encompassed in this study comprise air-cooling, water-cooling through tubes affixed to the rear side of the PV, cooling via the utilization of aluminum oxide-water nano-fluid in tubes attached to the back of the PVs, and cooling through water spraying on the front PV.

 The experiments were conducted over four consecutive days, utilizing the four proposed models under consistent weather conditions. Each day, an experimental PV equipped with a cooling method was tested alongside a PV lacking any cooling mechanism. The experiments were carried out between the hours of 8 am and 5 pm on July 3rd, 4th, 5th, and 6th in the city of Jeddah, located in Saudi Arabia. The temperature sensor has recorded and documented the surface temperature of the PVs and the ambient air temperature at regular intervals throughout the day. To quantify the surface temperature of the PV, a sensor is strategically positioned at the midpoint of the rectangular configuration of the PV. In conjunction with hourly temperature measurements, the solar meter was employed to measure and record the intensity of solar radiation. Additionally, the voltage and current generated by the PV were measured at the onset of each hour to quantify the power output during that hour.

**Table 3. Measuring equipment.**

| Equipment | Range Accuracy |
|---|---|
| **Solar meter**<br>**DT-1307 CEM** | **Up to 1999 W/m2**<br>**+/- 10 W/m2** |
| **Multimeter**<br>**APPA 72** | **Up to 10 (A)**<br>**Up to 1000 (V)**<br>**+/- 1%** |
| **Temperature sensor (NTC 100)** | **-5–125 (˚C)**<br>**+/- 2%** |

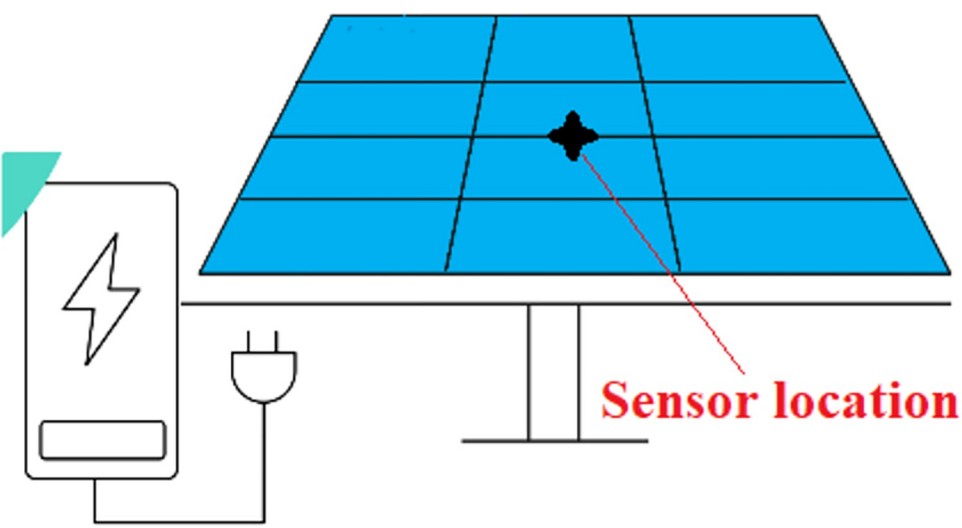

**Fig 4. Sensor location.**

## 2.4. Calculating the efficiency of a photovoltaic panel

To assess the operational efficacy of PVs, it is imperative to monitor not only their surface temperature but also their efficiency on an hourly basis. The efficiency of a PV can be calculated using the following method [45]:

**A: Short-Circuit Current ($I_{sc}$) and Open-Circuit Voltage ($V_{oc}$) Measurements:.** The short-circuit current ($I_{sc}$) and open-circuit voltage ($V_{oc}$) of the PV panel are measured using a multimeter. The short-circuit current is measured by connecting the multimeter to the panel's terminals and recording the current when the panel is short-circuited. The open-circuit voltage is measured by connecting the multimeter to the panel's terminals and recording the voltage when the panel is open-circuited.

**B: Maximum Power Point (MPP) Measurements.** The maximum power point (MPP) current ($I_{MPP}$) and voltage ($V_{MPP}$) are measured using a variable load or a maximum power point tracker (MPPT). The MPP is the point at which the panel produces its maximum power output.

$$P_{MAX} = V_{MPP} \times I_{MPP} \tag{1}$$

**C: Fill Factor (FF) Calculation.** The fill factor (FF) is calculated using the measured values of $I_{sc}$, $V_{oc}$, and $P_{MAX}$:

$$\mathrm{FF} = (P_{MAX})/(V_{oc} \times I_{sc}) \tag{2}$$

**D: Efficiency (η) Calculation.** The efficiency (η) of the PV panel is calculated using the fill factor, maximum power point values, and the irradiance (I) and area (A) of the panel:

$$\eta = (V_{oc} \times I_{sc} \times ff)/(A \times I) \tag{3}$$

## 2.5. Uncertainty analysis

Experiment-obtained data were subjected to uncertainty analysis to determine the degree of measurement uncertainty. Type A uncertainty was associated with the inaccuracy of the measuring instruments, which was derived from the devices' technical data documents. Type B uncertainty included random variations in measurements, which were determined by the

repeatability of investigations. Using the error propagation technique, the total uncertainty was then calculated [56]. The results indicated that the uncertainties associated with measuring solar radiation intensity, ambient temperature, and PV surface temperature were 2.3%, 1.2%, and 1.7%, respectively. In addition, the uncertainty associated with quantifying the output voltage and current of the PVs was estimated to be 0.8% and 1.1%, respectively. Given the minor uncertainties obtained, it is possible to conclude that the accuracy of the measurements was adequate.

## 3. Experimental results and discussion

The experiment was conducted in accordance with the protocol described earlier, with the goal of evaluating the operational efficiency of PVs using four distinct cooling methods. A solar meter module was employed to measure and record the solar radiation throughout the day. The data presented in Fig 5 shows that the radiation intensity varies quantitatively with the duration of daylight hours. As expected, the solarimeter recorded the highest level of radiation intensity at noon, due to the nearly perpendicular angle of the solar radiation with the Earth's surface. The peak value observed was approximately 930 watts per square meter, occurring at around 13:00.

The experiment was conducted over four consecutive days under consistent weather conditions, with each day dedicated to a distinct cooling system. As shown in Fig 5, the radiation intensity remained relatively consistent during the same hours across all four days. Given the

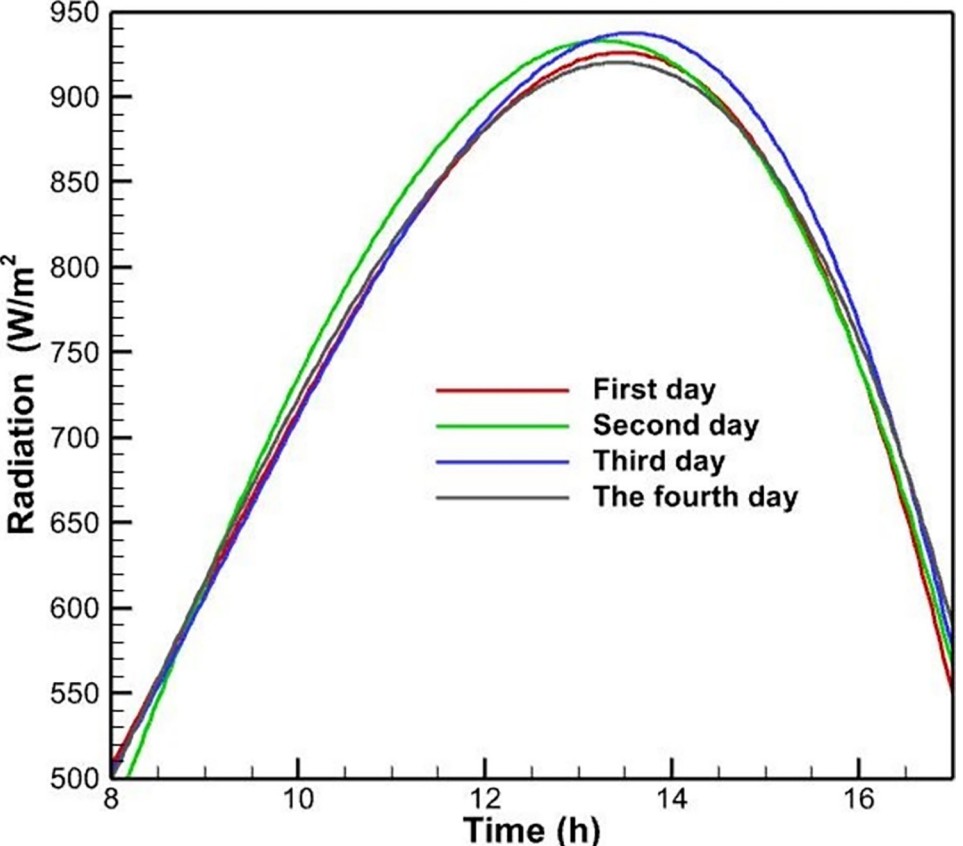

**Fig 5. Intensity of solar radiation at various hours of the day on four distinct days.**

identical meteorological conditions, the results can be considered a reliable basis for comparing the efficacy of the various cooling techniques.

The findings of this study have the potential to inform researchers and manufacturers on how to enhance the efficiency of PVs. Additionally, they provide valuable insights for selecting the most suitable cooling technique during the installation and operation of these PVs.

Initially, PVs were studied without a cooling system. According to Fig 6, as the morning progresses and noon approaches, the surface temperature of the PV rises and reaches a maximum of 52.5 degrees Celsius as a result of the intensification of solar radiation. This increase in surface temperature is due to the absorption of solar energy by the PV cells, which generates heat and raises the operating temperature of the system. As seen in Fig 7, the efficiency of PVs has an inverse relationship with their temperature; as the temperature rises, the efficiency of the PVs decreases. This phenomenon is well-established in the literature and can be attributed to the temperature-dependent characteristics of semiconductor materials used in PV cells. Specifically, as the temperature increases, the bandgap energy of the semiconductor material decreases, leading to a reduction in the open-circuit voltage and, consequently, the overall electrical efficiency of the PV system. Midday is when the efficacy of the PVs without a cooling system reaches its lowest level, which is 12 percent, due to the peak in solar radiation and the resulting high operating temperature.

In addition to the test conducted without a cooling system, another PV was evaluated using an air-cooling method. The fan was utilized to propel air onto the heatsinks positioned behind

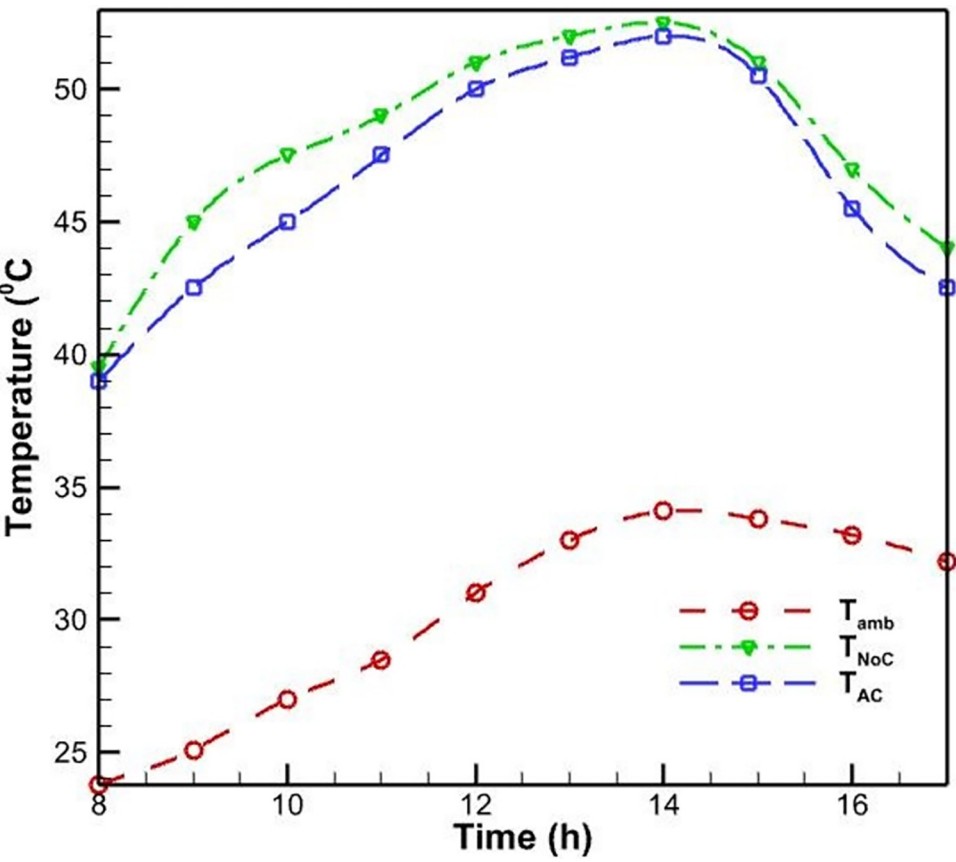

**Fig 6. Temperature of the ambient, PV with forced air-cooling method, and PV without cooling system during the day.**

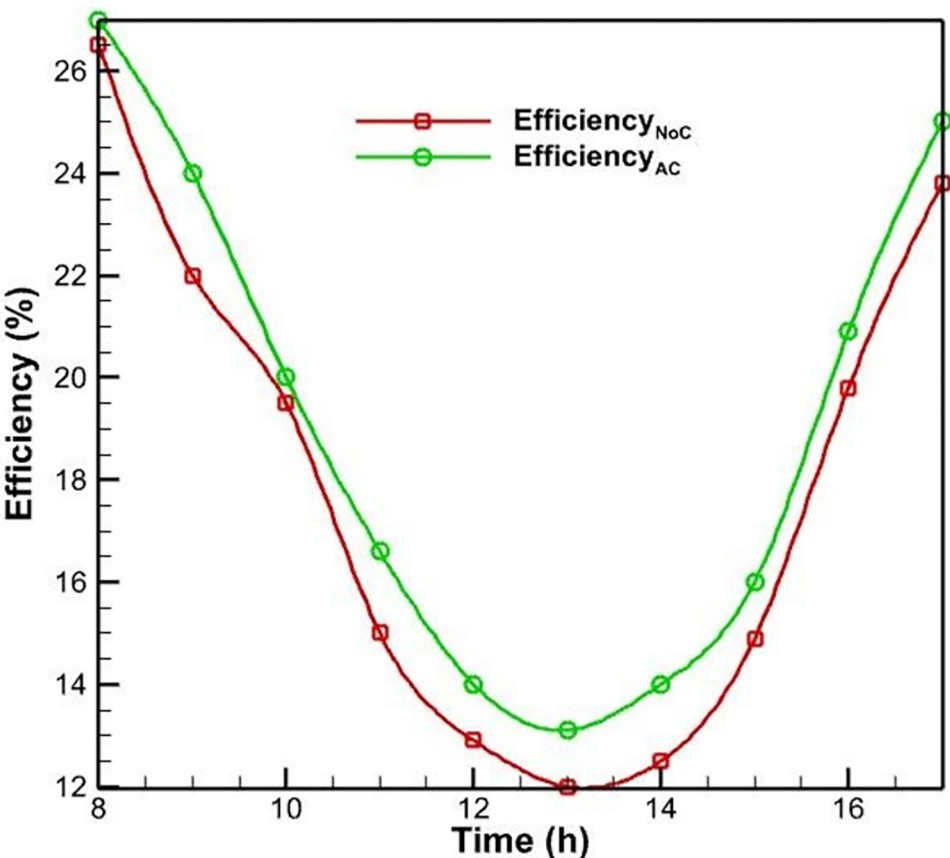

**Fig 7. The PV efficiency with forced air-cooling and without a cooling system.**

the PVs. Fig 6 demonstrates that the application of forced convection heat transfer results in a slight reduction in the surface temperature of the PV system, as compared to its temperature when no cooling system is present. Nevertheless, the rate of temperature change remains constant. This similarity can be explained by the fact that as daytime radiation increases, so does the environment's temperature; consequently, cooling has little influence on temperature fluctuations. In this instance, the PV's maximum temperature reaches 51 degrees Celsius, which is lower than the PV's maximum temperature without a cooling system. In this cooling method, the process of changing the efficiency of the PVs is identical to the case without cooling. In this case, the lowest PV efficiency occurs at noon and is equal to 13.1, resulting in a 1.1% increase in efficiency compared to the case without cooling.

The limited effectiveness of the air-cooling method can be attributed to the relatively low heat transfer coefficient of air compared to water, which is the primary working fluid in more efficient cooling techniques like the water-spray cooling system. The air-cooling system is unable to dissipate the heat generated within the PV module as effectively as the water-spray cooling, leading to a smaller improvement in the overall electrical efficiency (see Fig 7).

Fig 8 is a 3D plot that illustrates the relationship between the temperature and efficiency of a PV panel in the forced-air-cooling mode throughout the day. The plot shows that as the day progresses and midday approaches, the temperature of the PV panel increases, which in turn causes the panel's efficiency to decrease. The plot also reveals that the temperature reaches its maximum value around noon, corresponding to the lowest efficiency value. This plot is a useful tool for understanding the thermal behavior of PV panels and how it affects their efficiency.

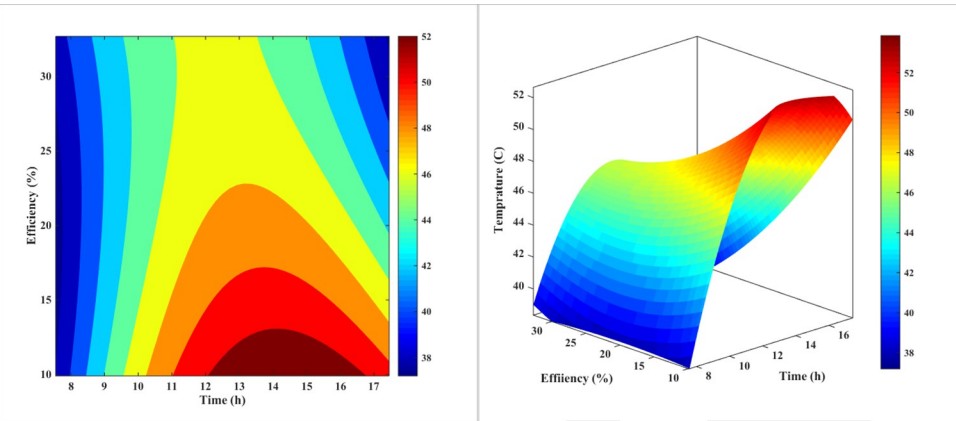

**Fig 8. The 3D diagram of PV surface temperature and its efficiency during different hours of the day for air-cooling method.**

The results suggest that the cooling system is effective in reducing the temperature of the PV panel, but it is not sufficient to maintain a constant efficiency throughout the day. The plot also highlights the importance of considering the thermal behavior of PV panels when designing and optimizing their performance.

The plot can be used to identify the optimal operating conditions for the PV panel, such as the temperature and efficiency at which the panel operates. It can also be used to design and optimize the cooling system to improve the efficiency of the PV panel. Overall, Fig 8 provides a valuable insight into the thermal behavior of PV panels and how it affects their efficiency.

The PV was tested with a water-cooling method on the rear the following day. In this instance, water flows through the pipes that are affixed to the back of the PVs. According to Fig 9, the temperature of the PV rises throughout the day and reaches its peak between 12:00 and 14:00, when it reaches approximately 46 degrees Celsius. This is 9% lower than the maximum temperature observed in the air-cooling method, suggesting that the water-cooling method is more efficient in decreasing temperature. This can be attributed to the higher specific heat capacity and thermal conductivity of water compared to air. The specific heat capacity of water (4.18 kJ/kg·˚C) is significantly higher than that of air (1.005 kJ/kg·˚C), allowing water to absorb more heat per unit mass for the same temperature rise. Additionally, the thermal conductivity of water (0.6 W/m·˚C) is about 24 times higher than that of air (0.025 W/m·˚C), facilitating more efficient heat transfer from the PV surface to the cooling medium. Moreover, the temperature of the water is largely independent of the temperature of the air around it; in fact, the water temperature rises less rapidly as the air temperature increases.

Upon evaluating the efficiency of the PV system under water-cooling conditions, the findings demonstrate a substantial increase in efficiency compared to the system's performance under air-cooling conditions. In the lowest mode, the efficiency value reaches 14.1%, representing a 7.6 and 17.5% improvement in performance over an air-cooling method and no cooling system, respectively (see Fig 10). This significant enhancement in efficiency can be attributed to the more effective heat dissipation achieved by the water-cooling system. By maintaining a lower operating temperature, the water-cooling method helps to minimize the temperature-induced losses in the PV's open-circuit voltage and fill factor, which are the primary contributors to the overall efficiency reduction at higher temperatures.

Fig 11 demonstrates qualitatively the impact of temperature variations on the PV's performance in cooling mode with water on the rear during the day. Contrary to what is depicted in Fig 8 for air-cooling, water-cooling is a more regular process than air-cooling. As can be seen,

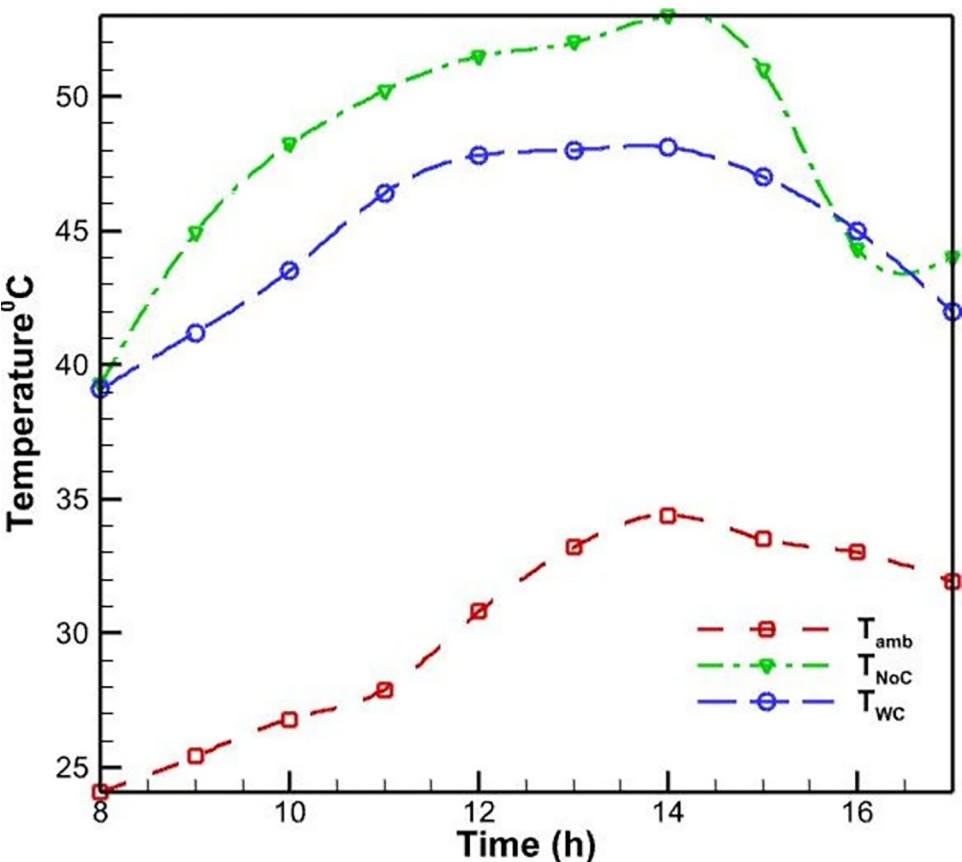

**Fig 9. The ambient temperature, the PV throughout the day without a cooling system, and the PV with a water-cooling system on the back.**

as the day progresses and noon approaches, the temperature of the solar PV rises, and its efficacy decreases as a result. In contrast to the process of chilling with air, this temperature increase reaches its peak around 2:00 p.m., when the efficiency value is at its lowest point. This more consistent behavior of the water-cooling system can be attributed to the thermal inertia of the water, which helps to dampen the rapid temperature fluctuations observed in the air-cooling method.

The superior performance of the water-cooling method can be attributed to the enhanced heat transfer capabilities of water compared to air. The water flowing through the pipes behind the PV panel effectively dissipates the heat generated within the system, maintaining a lower operating temperature and, consequently, a higher electrical efficiency. This finding highlights the importance of selecting the appropriate cooling technique to optimize the performance of PV systems, with water-cooling emerging as a more effective solution compared to air-cooling.

On the third day, the second day's test was repeated under the same conditions, with the exception that the working fluid was replaced with an aluminum oxide-water nanofluid. According to Fig 12, the use of nanofluid diminishes the magnitude of daily temperature fluctuations. In contrast to the previous two cooling modes, the PV's temperature rises gradually at first and achieves its maximum value, 45.2 degrees Celsius, between 11:00 and 14:00. This maximum temperature is 1.7% lower than the water-cooling method mode's maximum temperature.

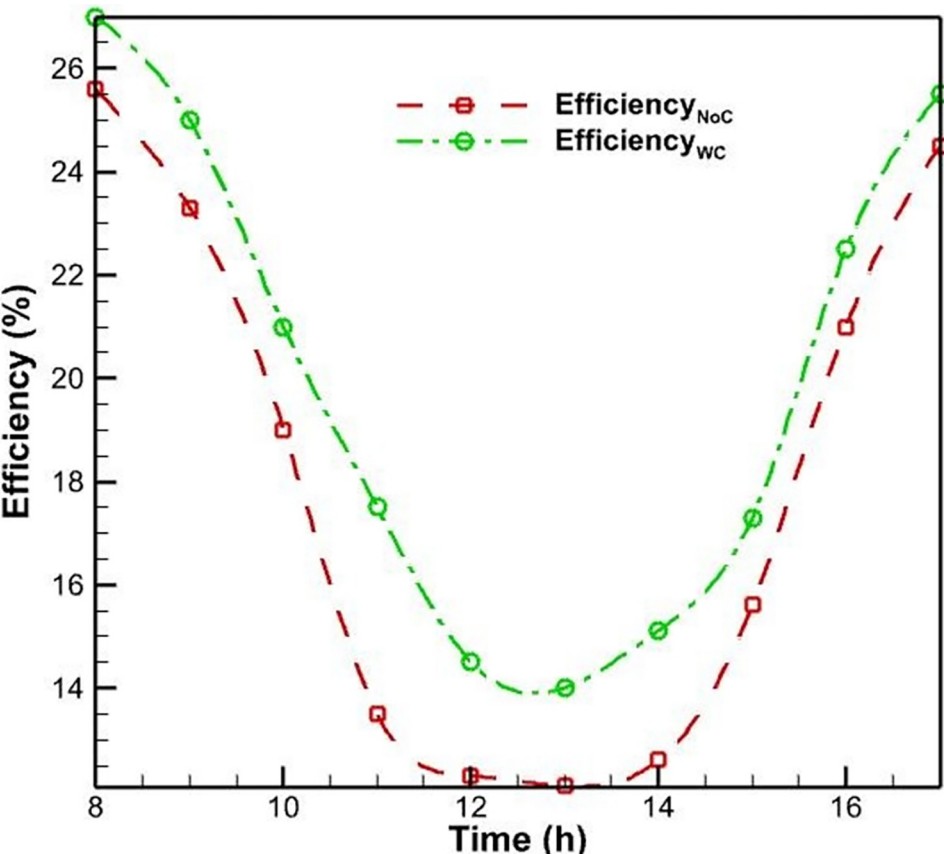

**Fig 10. The PV efficiency with forced water-cooling and without a cooling system.**

The improved temperature regulation with the nanofluid cooling can be attributed to the enhanced thermal properties of the nanofluid compared to pure water. The addition of aluminum oxide nanoparticles (typically with a size range of 10–50 nm) to the water increases the thermal conductivity of the fluid, facilitating more efficient heat transfer from the PV surface.

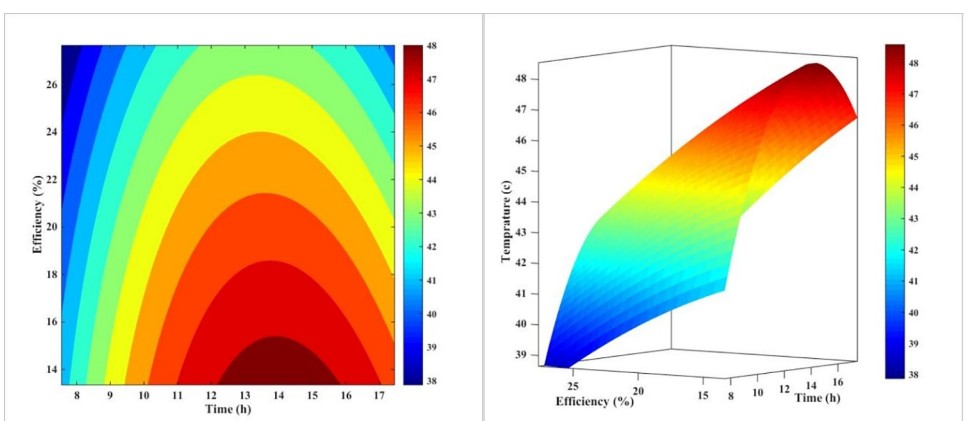

**Fig 11. The 3D diagram of PV surface temperature and its efficiency during different hours of the day for the water-cooling method on the rear.**

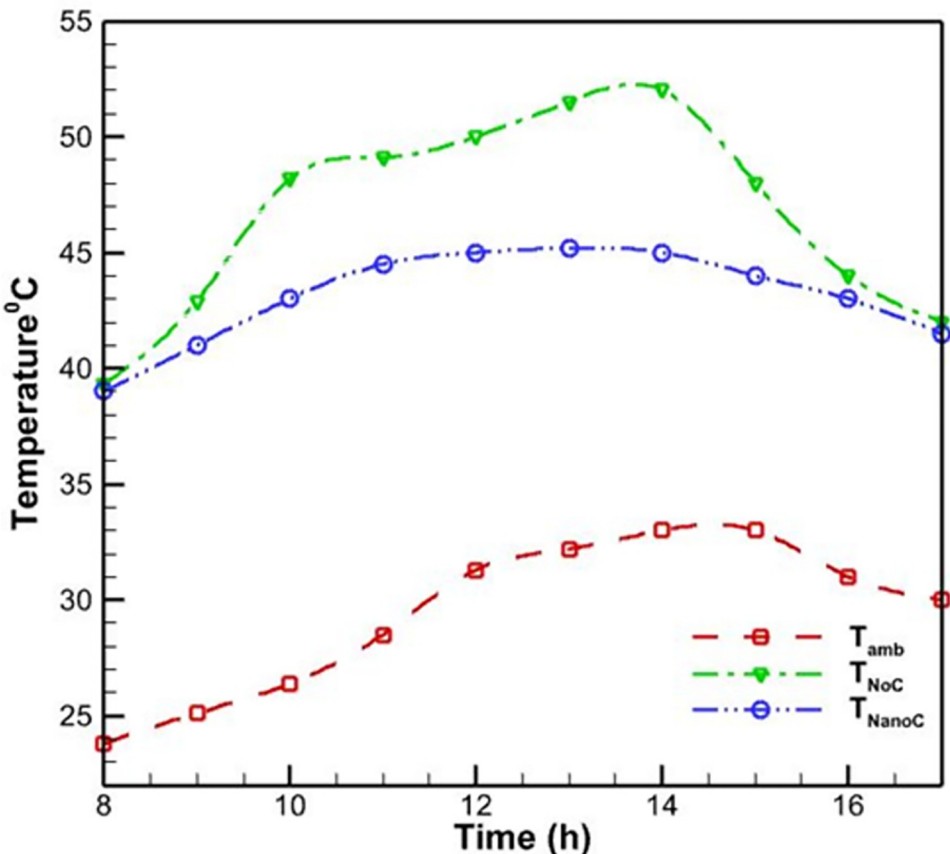

**Fig 12. Temperature of the ambient, PV with nanofluid cooling method on the rear, and PV without cooling system during the day.**

The nanoparticles act as "thermal bridges," enhancing the overall heat transfer coefficient of the nanofluid and allowing for faster heat dissipation. Additionally, the nanoparticles can alter the fluid flow characteristics, promoting better mixing and heat distribution within the cooling system.

As expected, Fig 13 demonstrates that the efficiency of the PVs in this mode has considerably increased compared to previous modes. At 12:30, the lowest efficacy of the PV is measured, which is 15.2%. Compared to the water-cooling mode, this value improved performance by 7.8% in the worst circumstances. By adding nanoparticles of aluminum oxide to water, the water's overall heat transfer coefficient increases, and heat transfer occurs at a faster rate.

Fig 14 depicts qualitatively the effect of the PV's daytime temperature variations on its cooling mode efficiency with nanofluid. In contrast to what was observed in Figs 8 and 11, nanofluid cooling is more regular than air and water-cooling. In addition, as the day progresses and noon approaches, the temperature of the PV rises, and as the temperature rises, the PV's efficacy decreases. This temperature increase reaches its maximum value at approximately 1:00 p. m. when the efficiency value is at its lowest. This is in contrast to the utmost temperature for air-cooling mode at noon and water-cooling mode at two p.m. This result indicates that adding nanoparticles to water modifies the PV's efficacy.

The more consistent and gradual temperature profile observed with the nanofluid cooling can be attributed to the enhanced thermal inertia of the nanofluid compared to pure water.

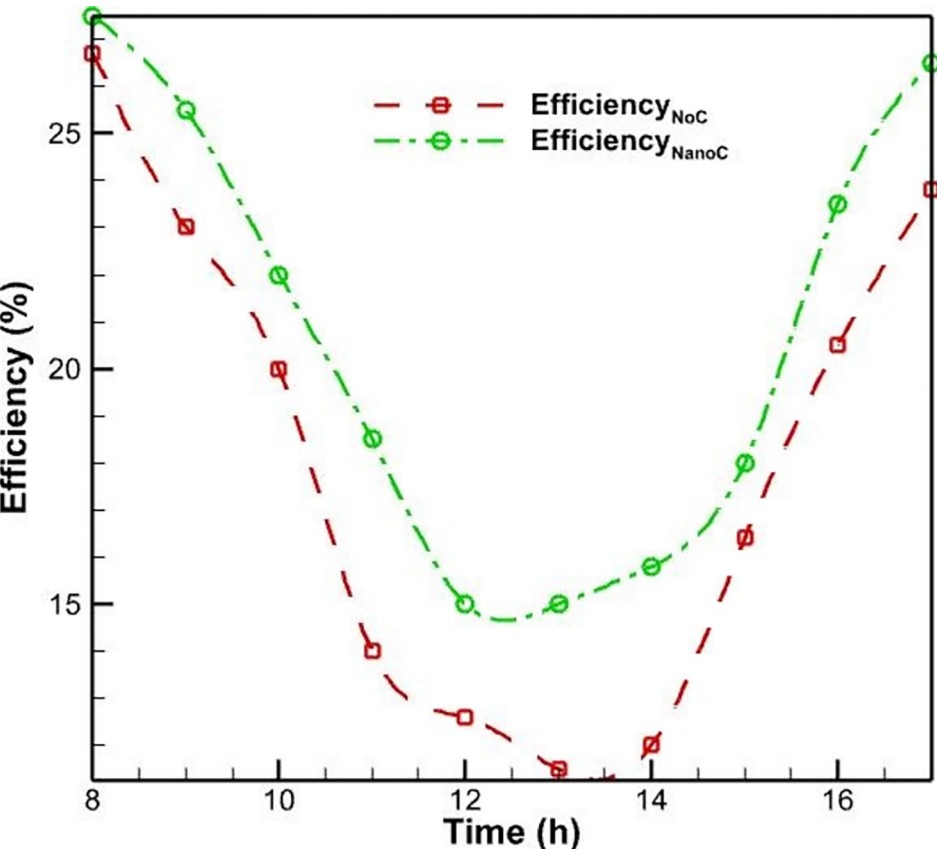

**Fig 13. The PV efficiency with nanofluid cooling and without a cooling system.**

The presence of the nanoparticles increases the effective heat capacity of the cooling medium, allowing it to absorb and dissipate heat more effectively and smoothing out the temperature fluctuations throughout the day. This improved thermal management contributes to the higher and more stable PV efficiency achieved with the nanofluid cooling system.

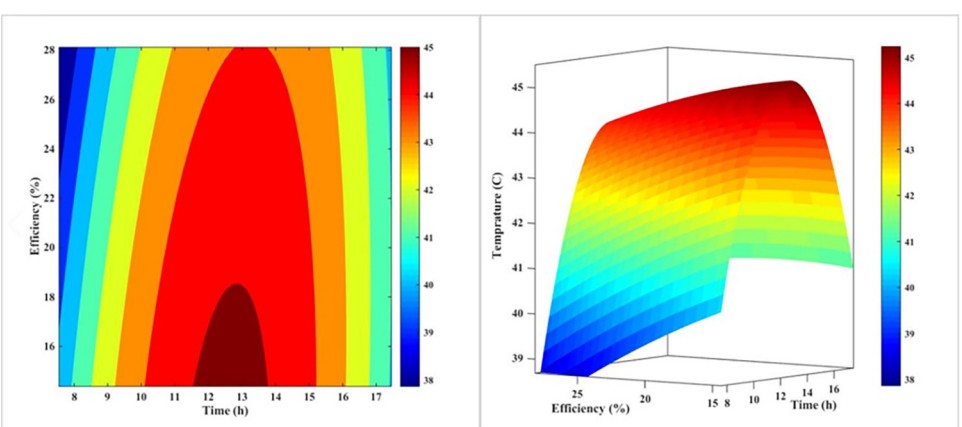

**Fig 14. The 3D diagram of PV surface temperature and its efficiency during different hours of the day for the nanofluid cooling method on the rear.**

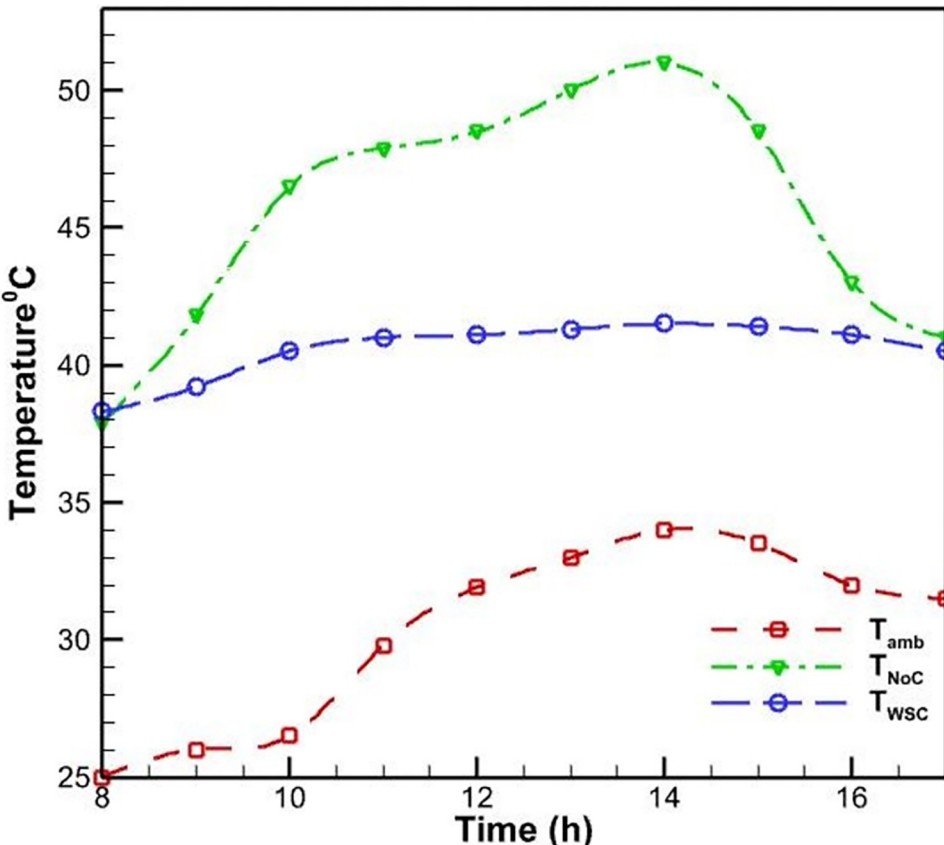

**Fig 15. Ambient temperature, PV without a cooling system throughout the day, and PV with a water-cooling method from the front surface (spraying water).**

On the fourth day, the PV was tested by sprinkling water on the opposite side to cool it. Due to the proximity of water molecules sprayed directly onto the PV cells in this method, heat transfer occurs at a faster rate. In addition, owing to the structure of this method, the overall heat transfer coefficient is significantly higher than in previous methods.

As anticipated, according to Fig 15 the daytime temperature trend of the PV is nearly constant and reaches a maximum of 41 degrees Celsius. In this instance, the maximum temperature was 21.9%, 19.6%, 10.8%, and 9% cooler than with no cooling, air-cooling, water on the back, and nanofluid cooling, respectively. These results indicate that sprinkling water directly from the front onto the surface of PV cells is more effective than other methods at lowering its temperature. The direct contact between the water droplets and the PV surface allows for more efficient heat dissipation, as the water can directly absorb the heat generated within the PV cells, rather than relying on conductive or convective heat transfer through intermediate materials or structures.

Also, according to Fig 16, the lowest PV efficiency in this instance is 16.4%, which is greater than the other cooling methods. This significant improvement in efficiency can be attributed to the more effective temperature regulation achieved by the water-spray cooling. By maintaining the PV's operating temperature at a lower and more consistent level, the water-spray method helps to minimize the temperature-induced losses in the PV's electrical parameters, such as the open-circuit voltage and fill factor, which are the primary contributors to the overall efficiency reduction at higher temperatures.

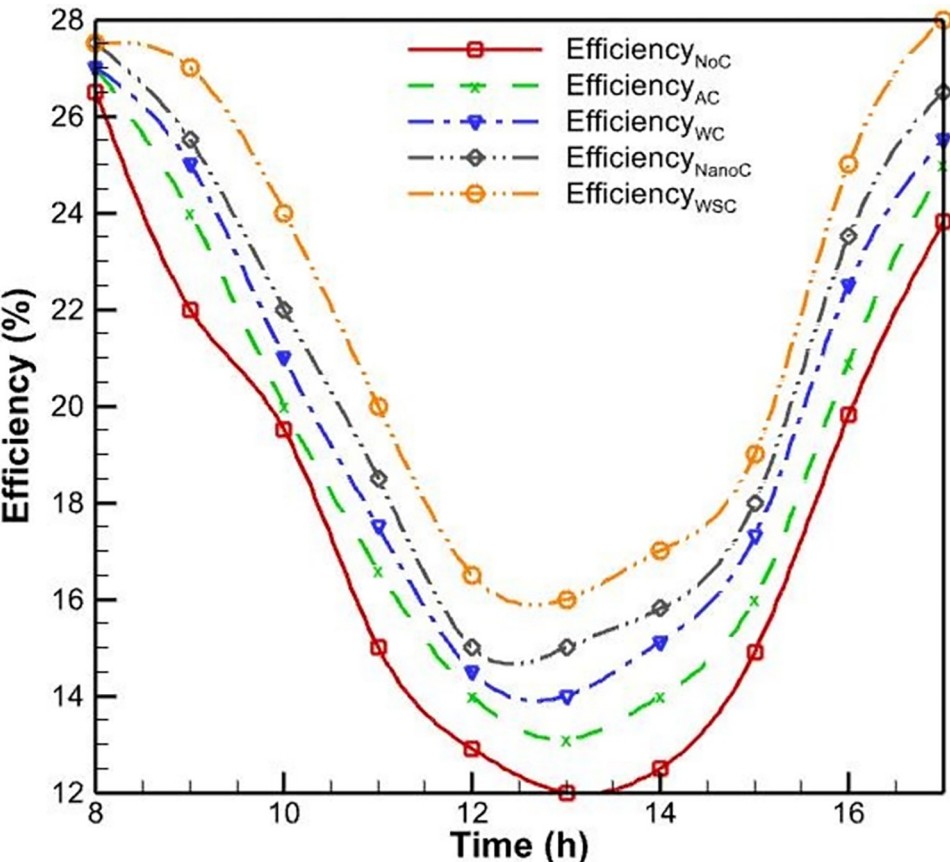

**Fig 16. PV efficiency with cooling and without cooling systems.**

Fig 17 illustrates the impact of temperature fluctuations on the efficiency of a PV system throughout the day, specifically when water is sprayed on the front surface of the PV. In this case, the range of variations in efficiency and temperature is much narrower than with other methods. The fact that the contour lines in this method are nearly vertical at various times

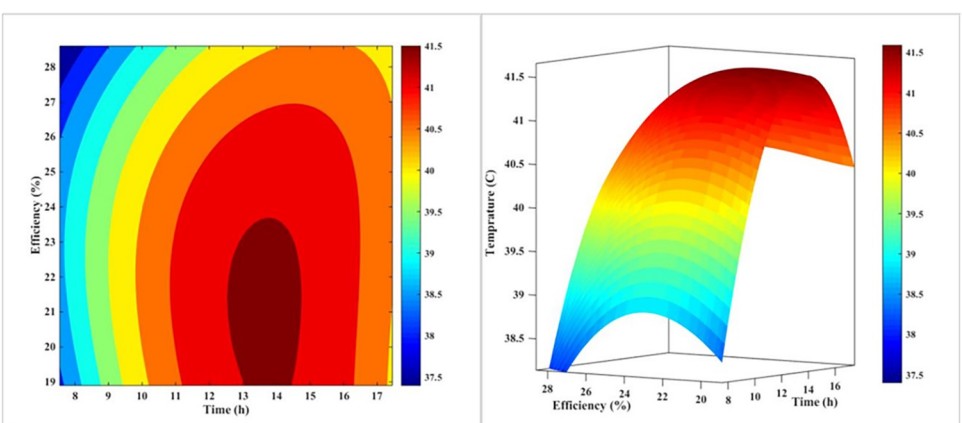

**Fig 17. The 3D diagram of PV surface temperature and its efficiency during different hours of the day for water-cooling method from the front surface (sprinkling water).**

demonstrates its superiority over other cooling techniques. This consistent performance can be attributed to the efficient heat dissipation and the ability of the water-spray cooling to maintain a more stable operating temperature for the PV system, even as the environmental conditions change throughout the day.

The water-spray cooling method emerges as the most effective approach among the techniques investigated in this study. The direct contact between the water droplets and the PV surface, coupled with the high overall heat transfer coefficient, enables a significant reduction in the PV's operating temperature and a substantial improvement in its electrical efficiency. This finding highlights the potential of the water-spray cooling system as a cost-effective and practical solution for enhancing the performance of PV systems in real-world applications.

### 3.1. Comparison of methods

As previously mentioned, the experimental results were analyzed to investigate the performance of the PV in non-cooling and cooling modes using four other methodologies. According to Fig 18, the lowest temperature of the PV is approximately 39 degrees Celsius in all circumstances, which corresponds to the early morning hours. The maximal temperature without cooling is 52.5 degrees Celsius, which surpasses the elevation of other methods. The maximal temperature of the PV when cooled by air is 51 degrees Celsius, which is a very poor performance when compared to the case without cooling. The performance of cooling the PV with water on the back and aluminum oxide-water nanofluid was significantly superior to cooling with air, with the maximal temperature of the PV reaching 46 and 45,2 degrees Celsius, respectively. Meanwhile, nanofluid cooling has superior temperature-lowering capabilities.

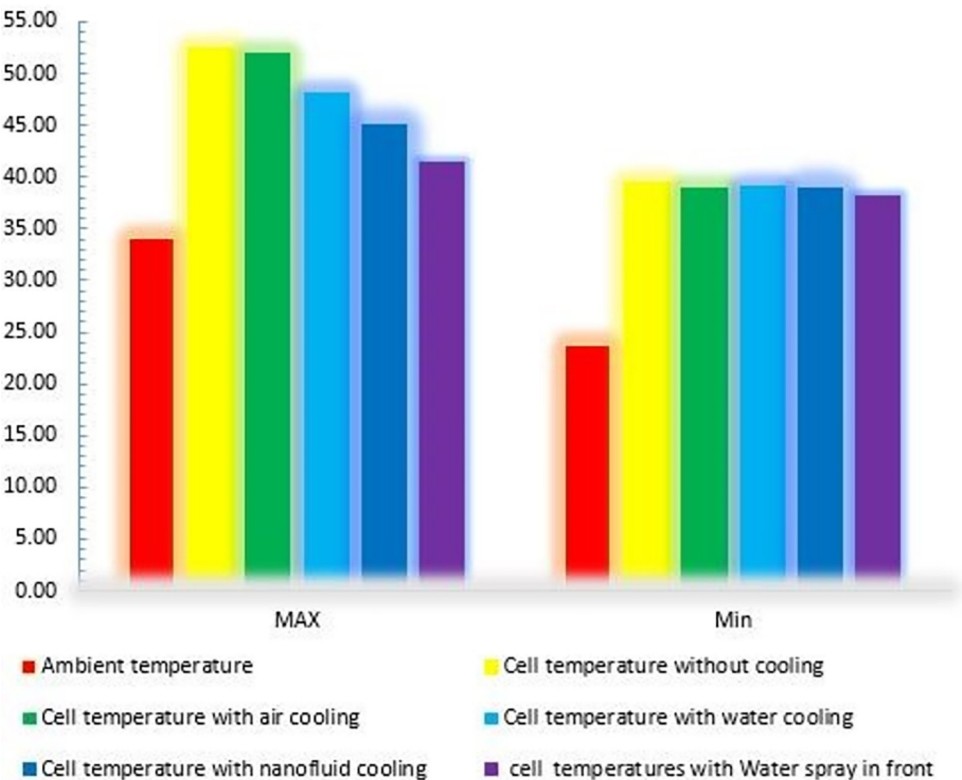

**Fig 18. The maximum and minimum temperature of the PV during the day.**

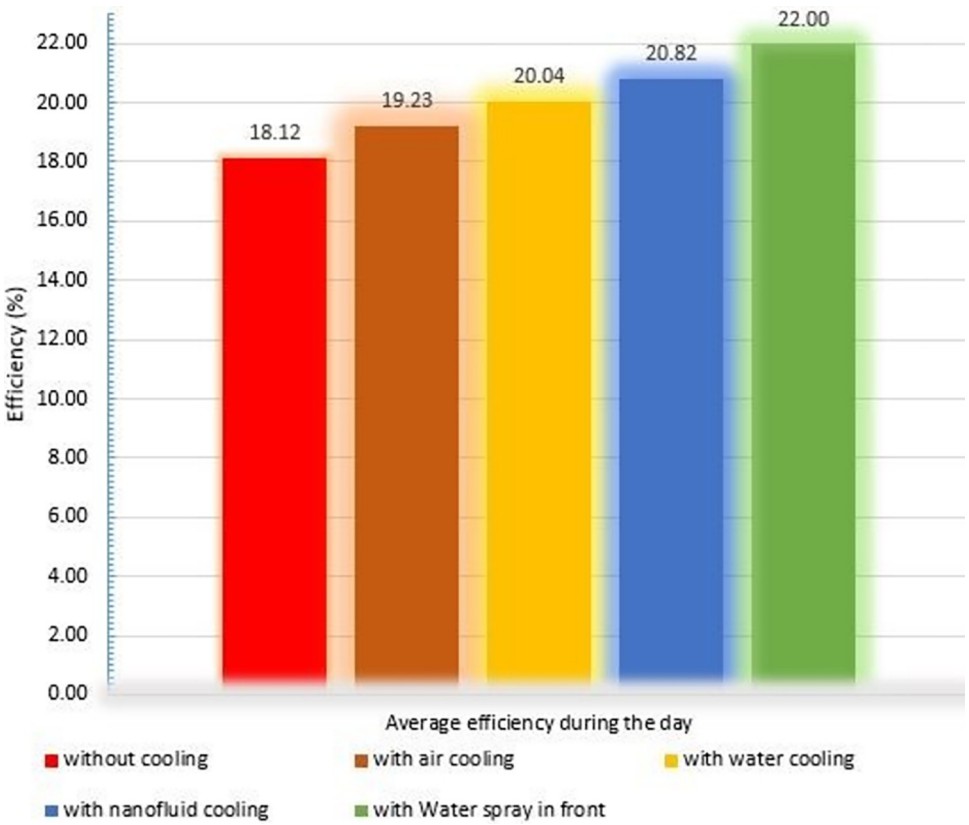

**Fig 19. Bar graph of the average efficiency of PVs during the day.**

Finally, the water spray on the PV's front surface was tested, and the results demonstrated that this method is significantly superior to the previous three cooling methods and can set the temperature to a maximum of 41 degrees Celsius.

In Fig 19, the average efficiency of PVs during the day for various cooling techniques is compared. According to this figure, the average efficacy of the PV in the cooling method with water spraying on the front of the PV is 22%, resulting in a performance increase of 3.9% compared to the case without a cooling system.

The average daytime efficiency of air, water in the back, and nanofluid cooling methods was 19.23%, 20.04%, and 20.83%, respectively, which is an improvement of 1.1%, 1.9%, and 2.7% over the no cooling system. In other words, according to Fig 20, the air-cooling method increases the efficiency of PVs by 6.1%, while the cooling methods with water on the back, nanofluid, and water spraying on the front of the PV increase the efficiency by 10.59%, 14.90%, and 21.44%, respectively.

## 4. Conclusion

This experimental study investigated the impact of different cooling methods on the electrical efficiency of PV. Four cooling techniques were evaluated, including air, water at the back of the panel, aluminum oxide-water nanofluid at the back of the panel, and water-spraying on the front surface of the panel.

The results showed that the water-spraying method on the front surface of the PV panel was the most effective cooling approach. This method resulted in a 21.9% reduction in the

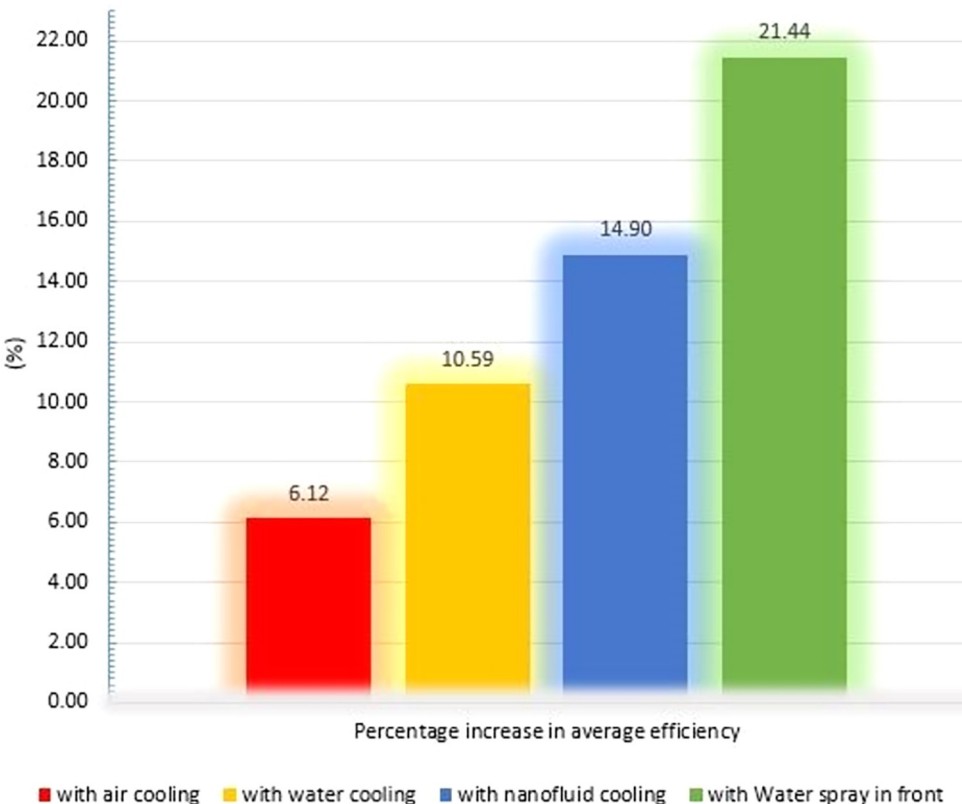

**Fig 20. Bar graph of increasing efficiency of PVs during the day compared to the state without cooling.**

maximum panel temperature compared to the non-cooling condition, and increased the average daily efficiency of the panel to 22%, which is a 3.9% improvement over the non-cooling case. In comparison, the air-cooling, water-cooling at the back, and nanofluid-cooling at the back methods showed 1.1%, 1.9%, and 2.7% efficiency improvements, respectively.

These findings indicate that water-spraying cooling on the front surface of PV panels is a cost-effective and efficient method for enhancing the performance of these panels. This technique can be considered as a practical solution for improving the performance of photovoltaic systems in real-world applications.

## 4.1. Recommendations

- Conduct further studies to investigate the impact of water-spraying system design parameters, such as water flow rate, spray angle, and spray pattern, on PV panel efficiency.

- Evaluate the long-term performance of the water-spraying cooling system and potential challenges related to corrosion, scaling, and freezing.

- Examine the economic and technical aspects of implementing the water-spraying cooling system at commercial and industrial scales.

These findings and recommendations can contribute to the development and optimization of photovoltaic technology towards renewable energy goals.

## Supporting information

**S1 Graphical abstract.**
(TIF)

## Author Contributions

**Conceptualization:** Ali Basem, Zakaria Mohamed Salem Elbarbary.

**Data curation:** Ali Basem, Zakaria Mohamed Salem Elbarbary, Natei Ermias Benti.

**Formal analysis:** Ali Basem, Azfarizal Mukhtar.

**Funding acquisition:** Zakaria Mohamed Salem Elbarbary.

**Investigation:** Ali Basem, Azfarizal Mukhtar, Natei Ermias Benti.

**Methodology:** Zakaria Mohamed Salem Elbarbary.

**Project administration:** Zakaria Mohamed Salem Elbarbary.

**Resources:** Ali Basem, Farruh Atamurotov, Natei Ermias Benti.

**Software:** Azfarizal Mukhtar, Farruh Atamurotov.

**Validation:** Azfarizal Mukhtar.

**Visualization:** Azfarizal Mukhtar, Farruh Atamurotov.

**Writing – original draft:** Farruh Atamurotov, Natei Ermias Benti.

**Writing – review & editing:** Natei Ermias Benti.

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
