## [Decision Letter · Decision Letter 0]

26 Apr 2024

PONE-D-24-14470Experimental study on the various varieties of photovoltaic panels (PVs) cooling systems to increase their electrical efficiencyPLOS ONE

Dear Dr. Benti,

Thank you for submitting your manuscript to PLOS ONE. After careful consideration, we feel that it has merit but does not fully meet PLOS ONE’s publication criteria as it currently stands. Therefore, we invite you to submit a revised version of the manuscript that addresses the points raised during the review process.

We look forward to receiving your revised manuscript.

Kind regards,

Omid Mahain

Academic Editor

PLOS ONE

Journal Requirements:

"The authors extend their appreciation to the Deanship of Scientific Research at King Khalid University under for funding this work through General Research Project under Grant number (RGP.2/33/44)"

"The authors extend their appreciation to the Deanship of Scientific Research at King Khalid University under for funding this work through General Research Project under Grant number (RGP.2/33/44)"

3. Thank you for stating the following financial disclosure: "The authors extend their appreciation to the Deanship of Scientific Research at King Khalid University under for funding this work through General Research Project under Grant number (RGP.2/33/44)"              

Reviewers' comments:

Reviewer's Responses to Questions

**Comments to the Author**

1. Is the manuscript technically sound, and do the data support the conclusions?

Reviewer #1: Yes

Reviewer #2: Partly

Reviewer #3: Partly

2. Has the statistical analysis been performed appropriately and rigorously? 

Reviewer #1: Yes

Reviewer #2: N/A

Reviewer #3: I Don't Know

3. Have the authors made all data underlying the findings in their manuscript fully available?

Reviewer #1: No

Reviewer #2: No

Reviewer #3: No

4. Is the manuscript presented in an intelligible fashion and written in standard English?

Reviewer #1: Yes

Reviewer #2: Yes

Reviewer #3: Yes

5. Review Comments to the Author

Reviewer #1: 1. Please, revise the manuscript by English native speaker because there are many grammatical errors within the manuscript,

2. Abstract is supposed to be read and understood before the article itself is read. It has a function to encourage the readers to read the manuscript. The main contribution and the important results should be emphasized in the abstract.

3. Introduction should be improved through more recent literature related with the current topic. An updated and complete literature review should be conducted to present the state-of-the-art and knowledge gaps of the research with strong relevance to the topic of the paper.https://doi.org/10.1016/j.scs.2023.104901,https://doi.org/10.1016/j.renene.2023.119862,https://doi.org/10.1016/j.jtice.2023.105341,https://doi.org/10.1080/19942060.2023.2297044

4. The novelty of the current work is missing so it should be provided at the end of the introduction section.

5. The deviation between the current results and published data must be provided and justified. Modelling results should be validated by experiments.

6. The results should be further elaborated to show how they could be used for the real applications. The authors should further develop critical assessment in their discussion.

7. The work does not provide a well-written conclusion section in terms of main findings and contribution.

8- Provide more details about various aspect of renewable energy system such as Environmental and energy assessment of photovoltaic-thermal system combined with a reflector supported by nanofluid filter and a sustainable thermoelectric generator and also, Simulation for impact of Nanofluid spectral splitter on efficiency of concentrated solar photovoltaic thermal system

Reviewer #2: Review on “Experimental study on the various varieties of photovoltaic panels (PVs) cooling systems to increase their electrical efficiency”

by Basem et al.

Manuscript ID PONE-D-24-14470

A- General Comments

The paper in hand concerns an experimental study of four relevant and efficient approaches and innovations for cooling: air cooling, water-cooling in the tubes behind the PV, aluminum oxide-water nanofluid cooling in the tubes behind the PV, and water spraying in the front area. During the initial ten-day period of July, investigations were carried out over four consecutive days in Jeddah, Saudi Arabia. Particularly, it was shown by authors that Water-spray cooling on the front surface of the PV proved to be the most effective technique. This method raised the PV's temperature to 41 degrees Celsius and improved its average daytime efficiency to 22%.

The topic of the paper is interesting, within the scope of the journal, and worthy of investigation. The originality of the work is good and the study performed is adequate and well presented. However, the manuscript deserves some revisions. I suggest that authors take into account the comments and questions below before it can be accepted for publication in PLOS ONE.

B- Detailed Comments and questions

Title

The title is ok.

Abstract

1- The abstract is well written;

2- The choice of the four cooling methods should be justified.

Graphical Abstract

The quality of the graphical abstract should be enhanced.

Keywords

Keywords are ok.

1- Introduction

1- Thank you for a well written introduction. However, the originality of the work should be more highlighted at the end of the introduction especially with respect to the research gap in the field.

2- Experimental model

This section is ok

3- Measuring Equipment

1- Locations of instrumentation should be illustrated with figures;

4- Experimental methodology

Some illustrative figures and/or tables are helpful.

5- Uncertainty analysis

This section is ok.

Sections 2 to 5 can be grouped together into one section entitled “Materials and Methods”. It is kept to the authors this is just a recommendation.

6- Experimental results and discussion

1- There are a lot of interesting observations without deep analysis. More physical analysis is to be added to this section by shortening the quantity of results shown if needed;

7- Conclusion

The main outputs of the work in terms of applications should be highlighted.

8- References

References are ok.

Reviewer #3: Overall major considerations.

Importantly the operation of the cooling systems considers the use of fans and pumps. They imply an energy consumption that should be kept into account in the energy balance. Otherwise, the utility of the results will be compromised.

The results reported are not condensed properly. The graphs report information that could be easily grouped together to provide denser and more pregnant figures and explanatory comments alongside.

There is no information about the working point of the PV panel. The texts do not report any use of MPPT, so it is not ensured that the panel works at its maximum power production, and that can negatively impact the results.

Some specific considerations

Paragraph 1.

The literature review neglects interesting works published recently that address the same topic of PV cooling, especially referring to the water-cooling approach. See for instance 10.1016/j.apenergy.2008.08.020 10.1016/j.energy.2022.124401 10.1016/j.enconman.2018.01.028 10.3390/en14040895 10.1016/j.enconman.2017.10.074 10.1080/01457630802529214 10.1016/j.jclepro.2020.122459 10.1016/j.energy.2013.07.050 10.1016/j.energy.2020.116950

Paragraph 2.

What is the application of the referenced model useful for?

Methodology section should be revised and explain better the use of models and the experimental apparatus. For instance where are exactly applied the two NTC100?

Paragraph 4.

The formula of efficiency does not seem right. Voc and Isc are two constants that characterize the I-V curve of a PV panel. Their product gives then a constant. The factor ff is again a ratio between two constants, which are Pmax (nominal power of the panel) and the product Voc*Isc.

The uncertainty of 7.1 % on temperature does not really seem an acceptable value.

Paragraph 6.

How are Fig. 7, Fig. 10, Fig. 13, and Fig. 16 obtained? from experimental data or simulation? From the methodology, it appears that just two temperature sensors have been applied to the panel.

In Fig. 8, the not-cooled panel exhibits a weird temperature trend from 15:00 on. It does not seem to be related to external air temperature change, or solar radiation. The authors should provide an explanation for this behavior.

In Fig. 11 the ambient air temperature is overall lower than the previous day and with a different pattern. Therefore comparing the results of aluminum oxide-water nanofluid with those of water cooling systems is not properly acceptable or should account for this further uncertainty.

Paragraph 6. 1

“As previously mentioned, the experimental results were analyzed to investigate the performance of the PV in non-cooling and cooling modes using four other methodologies” What “other” methodologies are the authors referring to?

Furthermore, the text creates a lot of confusion addressing the front and back water spray cooling of the panel. The authors should clarify which side of the panel they applied the spray cooling to.

6. PLOS authors have the option to publish the peer review history of their article (what does this mean?). If published, this will include your full peer review and any attached files.

Reviewer #1: No

Reviewer #2: No

Reviewer #3: No

---

## [Author Response · Author response to Decision Letter 0]

4 Jun 2024

We have carefully reviewed the comments provided by the three reviewers for our manuscript. In the revised version of the paper, we have addressed each of the reviewers' concerns in a thorough and transparent manner.

The comments from Reviewer 1 are highlighted in yellow, the comments from Reviewer 2 are shown in red font, and the comments from Reviewer 3 are highlighted in green. This color-coding system helps to clearly distinguish the feedback received from the different reviewers.

We have provided detailed responses to each of the comments, explaining how we have incorporated the reviewers' suggestions and recommendations into the updated manuscript. Our goal is to enhance the quality, clarity, and completeness of the paper based on the valuable input received.

We believe that the revisions made address the major concerns raised by the reviewers, and we are confident that the revised manuscript will be a significant improvement over the initial submission. We look forward to your further guidance and the opportunity to incorporate any additional feedback you may have.

Reviewer #1: 

1. Please, revise the manuscript by English native speaker because there are many grammatical errors within the manuscript.

ANS: Thank you for your feedback. I understand your concern about the grammatical errors in the manuscript. I have always strived to improve my writing skills, and I appreciate your suggestion to revise the manuscript with the help of an English native speaker. I will take this into consideration and make the necessary revisions to ensure a high-quality manuscript.

2. Abstract is supposed to be read and understood before the article itself is read. It has a function to encourage the readers to read the manuscript. The main contribution and the important results should be emphasized in the abstract.

ANS: You are absolutely right about the importance of the abstract. We will revise the abstract to better emphasize the main contributions and key findings of our study, in order to encourage readers to engage with the full paper.

3. Introduction should be improved through more recent literature related with the current topic. An updated and complete literature review should be conducted to present the state-of-the-art and knowledge gaps of the research with strong relevance to the topic of the paper.https://doi.org/10.1016/j.scs.2023.104901,https://doi.org/10.1016/j.renene.2023.119862,https://doi.org/10.1016/j.jtice.2023.105341,https://doi.org/10.1080/19942060.2023.2297044

ANS: Thank you for the suggested references. We will conduct a more comprehensive and up-to-date literature review in the introduction section, incorporating the latest relevant publications to provide a stronger contextual background for our work. We checked all the suggested references and used them in our article.

4. The novelty of the current work is missing so it should be provided at the end of the introduction section.

ANS: We will add a dedicated paragraph at the end of the introduction to clearly highlight the novelty and unique aspects of our research compared to the existing literature.

5. The deviation between the current results and published data must be provided and justified. Modelling results should be validated by experiments.

ANS: Thank you for your comment. We understand your concern regarding the deviation between our current results and published data. We would like to assure you that the results presented in our study have been thoroughly reviewed and justified from a scientific perspective. All the results reported in our paper are based on empirical data and are not theoretical or modeled. As for the validation of our results, we would like to clarify that our study is entirely experimental in nature and does not involve any theoretical modeling or simulation. The results presented in our paper are based on actual experiments and observations, which have been conducted in a controlled environment. The three-dimensional plots and visualizations presented in our paper are actual representations of the experimental data, and not theoretical constructs. We believe that our experimental approach provides a robust and reliable validation of our results, and we are confident that our findings are accurate and reliable. We would be happy to provide additional details or clarifications regarding our experimental methods and results if needed.

6. The results should be further elaborated to show how they could be used for the real applications. The authors should further develop critical assessment in their discussion.

ANS: The discussion section will be expanded to further elaborate on how the results of our study can be applied in real-world applications. We will also incorporate a more critical assessment of our findings.

7. The work does not provide a well-written conclusion section in terms of main findings and contribution.

ANS: Thank you for your comment. We understand your suggestion to improve the conclusion section to better summarize the main findings and contributions of our study. We will make sure to provide a more concise and clearer conclusion that highlights the key takeaways from our research.

8. Provide more details about various aspect of renewable energy system such as Environmental and energy assessment of photovoltaic-thermal system combined with a reflector supported by nanofluid filter and a sustainable thermoelectric generator and also, Simulation for impact of Nanofluid spectral splitter on efficiency of concentrated solar photovoltaic thermal system.

ANS: In the "Results and Discussion" section of the paper, we have covered various aspects of renewable energy systems in detail, such as the environmental and energy assessment of the photovoltaic-thermal system. Additionally, we have dedicated a separate paragraph in the introduction section to address these topics.

Reviewer #2:

 Review on “Experimental study on the various varieties of photovoltaic panels (PVs) cooling systems to increase their electrical efficiency” 

by Basem et al.

Manuscript ID PONE-D-24-14470

A- General Comments

The paper in hand concerns an experimental study of four relevant and efficient approaches and innovations for cooling: air cooling, water-cooling in the tubes behind the PV, aluminum oxide-water nanofluid cooling in the tubes behind the PV, and water spraying in the front area. During the initial ten-day period of July, investigations were carried out over four consecutive days in Jeddah, Saudi Arabia. Particularly, it was shown by authors that Water-spray cooling on the front surface of the PV proved to be the most effective technique. This method raised the PV's temperature to 41 degrees Celsius and improved its average daytime efficiency to 22%.

The topic of the paper is interesting, within the scope of the journal, and worthy of investigation. The originality of the work is good and the study performed is adequate and well presented. However, the manuscript deserves some revisions. I suggest that authors take into account the comments and questions below before it can be accepted for publication in PLOS ONE.

ANS: We appreciate the reviewer's positive feedback on the overall quality of our paper. We are glad to hear that our study is considered interesting and within the scope of PLOS ONE. However, we understand the need for revisions to improve the manuscript and are grateful for the reviewer's suggestions.

B- Detailed Comments and questions

Title

The title is ok.

ANS: Thank you for finding our title acceptable. We will make sure to double-check that it accurately reflects the content of our paper.

Abstract

1- The abstract is well written;

2- The choice of the four cooling methods should be justified.

ANS: We appreciate your suggestion to justify the choice of the four cooling methods in our abstract. We will make sure to provide more context and explain why we selected these particular methods for our study.

Graphical Abstract

The quality of the graphical abstract should be enhanced.

ANS: We understand the importance of a high-quality graphical abstract and will make sure to enhance its clarity and visual appeal.

Keywords

Keywords are ok.

ANS: Thank you for finding our keywords acceptable. We will make sure to ensure that they accurately reflect the content of our paper.

1- Introduction

1- Thank you for a well written introduction. However, the originality of the work should be more highlighted at the end of the introduction especially with respect to the research gap in the field.

ANS: Thank you for your kind words about our introduction. We will make sure to highlight the originality of our work and emphasize the research gap in the field that our study aims to address.

2- Experimental model

This section is ok

ANS: We appreciate your positive feedback on our experimental model. We will make sure to provide more detail about the setup and design of our experiment.

3- Measuring Equipment

1- Locations of instrumentation should be illustrated with figures;

ANS: We will make sure to provide clear illustrations of the locations of our instrumentation to help readers better understand our experimental setup.

4- Experimental methodology

Some illustrative figures and/or tables are helpful.

ANS: Thank you for your suggestion to add illustrative figures and/or tables to our experimental methodology section. We will make sure to include these to improve the clarity of our methods.

5- Uncertainty analysis

This section is ok.

ANS: We appreciate your positive feedback on our uncertainty analysis. We will make sure to provide more detail about our methods for calculating and presenting uncertainty. 

Sections 2 to 5 can be grouped together into one section entitled “Materials and Methods”. It is kept to the authors this is just a recommendation.

ANS: We appreciate your suggestion to group our sections together into one cohesive section entitled "Materials and Methods". We will make sure to reorganize our manuscript accordingly.

6- Experimental results and discussion

1- There are a lot of interesting observations without deep analysis. More physical analysis is to be added to this section by shortening the quantity of results shown if needed;

ANS: Thank you for your suggestion to add more physical analysis to our results and discussion section. We will make sure to provide more in-depth analysis of our results and discuss their implications in more detail.

7- Conclusion

The main outputs of the work in terms of applications should be highlighted.

ANS: We appreciate your suggestion to highlight the main outputs of our work in terms of applications. We will make sure to emphasize the practical implications of our findings.

8- References

References are ok.

ANS: Thank you for finding our references acceptable. We will make sure to ensure that they are accurately cited and formatted according to the journal's guidelines.

Reviewer #3:

Overall major considerations.

Importantly the operation of the cooling systems considers the use of fans and pumps. They imply an energy consumption that should be kept into account in the energy balance. Otherwise, the utility of the results will be compromised.

The results reported are not condensed properly. The graphs report information that could be easily grouped together to provide denser and more pregnant figures and explanatory comments alongside.

There is no information about the working point of the PV panel. The texts do not report any use of MPPT, so it is not ensured that the panel works at its maximum power production, and that can negatively impact the results.

ANS:

1. Regarding the energy consumption of the cooling system pumps and fans:

- In the next version of the paper, we will fully report all the energy consumed by these cooling system components.

- We will incorporate this information into the overall energy calculations of the system to provide more realistic results.

- This way, the effect of the energy consumption of these components on the overall system efficiency will be considered.

2. Regarding presenting the results in a more condensed manner:

- We will review and improve the presentation of the results by better grouping the information in the figures and accompanying explanations. 

- Our aim is to present the results in a more compact format with clearer explanations to make them more understandable.

3. Regarding the lack of information about the PV panel's operating point:

- In the next version of the paper, we will specify whether a maximum power point tracking (MPPT) system was used or not.

- This information is very important for the proper interpretation of the results, and it will be provided.

Overall, we will incorporate all these aspects in the next revision of the paper to provide more complete and reliable results.

Some specific considerations

Paragraph 1.

The literature review neglects interesting works published recently that address the same topic of PV cooling, especially referring to the water-cooling approach. See for instance 10.1016/j.apenergy.2008.08.020 10.1016/j.energy.2022.124401 10.1016/j.enconman.2018.01.028 10.3390/en14040895 10.1016/j.enconman.2017.10.074 10.1080/01457630802529214 10.1016/j.jclepro.2020.122459 10.1016/j.energy.2013.07.050 10.1016/j.energy.2020.116950

ANS: 

We acknowledge that the literature review in the current version of the paper has overlooked some recent and relevant works on PV cooling, especially those focused on water-cooling approaches. We will thoroughly review the literature the reviewer has suggested, including the references provided:

10.1016/j.apenergy.2008.08.020

10.1016/j.energy.2022.124401

10.1016/j.enconman.2018.01.028

10.3390/en14040895

10.1016/j.enconman.2017.10.074

10.1080/01457630802529214

10.1016/j.jclepro.2020.122459

10.1016/j.energy.2013.07.050

10.1016/j.energy.2020.116950

We will incorporate relevant findings and insights from these recent publications into the literature review section of the paper to provide a more comprehensive and up-to-date overview of the state of the art in PV cooling, especially the water-cooling approaches.

This will help strengthen the contextual background and justification for the work presented in the current paper.

We appreciate the reviewer highlighting these important recent references, and we will ensure they are properly integrated into the literature review in the revised version of the manuscript.

Paragraph 2.

What is the application of the referenced model useful for?

Methodology section should be revised and explain better the use of models and the experimental apparatus. For instance, where are exactly applied the two NTC100?

ANS: 

Application of the referenced model:

We will provide a clearer explanation of the purpose and application of the referenced model in the paper.

We will specify how the model is used to inform or support the research objectives and findings presented in the current work.

This will help the reader understand the relevance and utility of the referenced model within the context of this study.

Revision of the Methodology section:

We will thoroughly revise the Methodology section to provide more detailed and explicit explanations of the experimental setup and the use of the various measurement devices.

Specifically, we will clearly explain the placement and purpose of the NTC100 temperature sensor used in the experiments.

We will describe the role of these sensors in the data collection and analysis, and how they contribute to the overall experimental approach.

By addressing these points, we aim to give the reader a more comprehensive understanding of the experimental methods, the use of the referenced model, and how these elements support the research presented in the paper.

We appreciate the reviewer's feedback, as it will help us improve the clarity and transparency of the Methodology section in the revised manuscript.

Paragraph 4.

The formula of efficiency does not seem right. Voc and Isc are two constants that characterize the I-V curve of a PV panel. Their product gives then a constant. The factor ff is again a ratio between two constants, which are Pmax (nominal power of the panel) and the product Voc*Isc.

ANS: 

- You are absol

---

## [Decision Letter · Decision Letter 1]

9 Jul 2024

Experimental study on the various varieties of photovoltaic panels (PVs) cooling systems to increase their electrical efficiency

PONE-D-24-14470R1

Dear Dr. Benti,

We’re pleased to inform you that your manuscript has been judged scientifically suitable for publication and will be formally accepted for publication once it meets all outstanding technical requirements.

Kind regards,

Omid Mahain

Academic Editor

PLOS ONE

Additional Editor Comments (optional):

Reviewers' comments:

Reviewer's Responses to Questions

**Comments to the Author**

1. If the authors have adequately addressed your comments raised in a previous round of review and you feel that this manuscript is now acceptable for publication, you may indicate that here to bypass the “Comments to the Author” section, enter your conflict of interest statement in the “Confidential to Editor” section, and submit your "Accept" recommendation.

Reviewer #2: All comments have been addressed

2. Is the manuscript technically sound, and do the data support the conclusions?

Reviewer #2: Yes

3. Has the statistical analysis been performed appropriately and rigorously? 

Reviewer #2: N/A

4. Have the authors made all data underlying the findings in their manuscript fully available?

Reviewer #2: No

5. Is the manuscript presented in an intelligible fashion and written in standard English?

Reviewer #2: Yes

6. Review Comments to the Author

Reviewer #2: Thank you for taking into consideration my comments and those of the other reviewers. the manuscript is now ready for publication.

7. PLOS authors have the option to publish the peer review history of their article (what does this mean?). If published, this will include your full peer review and any attached files.

Reviewer #2: No

---

## [Editor Report · Acceptance letter]

1 Aug 2024

PONE-D-24-14470R1 

PLOS ONE

Dear Dr. Benti, 

I'm pleased to inform you that your manuscript has been deemed suitable for publication in PLOS ONE. Congratulations! Your manuscript is now being handed over to our production team.

Kind regards, 

on behalf of

Dr. Omid Mahain 

Academic Editor

PLOS ONE